# Bacterial co-culture with cell signaling translator and growth controller modules for autonomously regulated culture composition

Kristina Stephens[1,2], Maria Pozo[1], Chen-Yu Tsao[1,2], Pricila Hauk [1,2] & William E. Bentley [1,2]

Synthetic biology and metabolic engineering have expanded the possibilities for engineered cell-based systems. The addition of non-native biosynthetic and regulatory components can, however, overburden the reprogrammed cells. In order to avoid metabolic overload, an emerging area of focus is on engineering consortia, wherein cell subpopulations work together to carry out a desired function. This strategy requires regulation of the cell populations. Here, we design a synthetic co-culture controller consisting of cell-based signal translator and growth-controller modules that, when implemented, provide for autonomous regulation of the consortia composition. The system co-opts the orthogonal autoinducer AI-1 and AI-2 cell-cell signaling mechanisms of bacterial quorum sensing (QS) to enable cross-talk between strains and a QS signal-controlled growth rate controller to modulate relative population densities. We further develop a simple mathematical model that enables cell and system design for autonomous closed-loop control of population trajectories.

[1] Fischell Department of Bioengineering, University of Maryland, 8278 Paint Branch Drive, 5102 Clark Hall, College Park, MD 20742, USA. [2] Institute for Bioscience and Biotechnology Research, University of Maryland, 8278 Paint Branch Drive, 5102 Clark Hall, College Park, MD 20742, USA. Correspondence and requests for materials should be addressed to W.E.B. (email: bentley@umd.edu)

Advances in synthetic biology and metabolic engineering have expanded the potential for engineered cell-based systems[1–3]. Engineered microbes enable environmentally friendly manufacture of valuable molecular products[4]. Also, smart bacteria have appeared that sense their environments and execute desired functions such as the synthesis and delivery of therapeutics[1,5–8]. It is well recognized, however, that engineering cells to carry out multiple functions or produce products through extensive interconnected pathways leads to new challenges. These include bottlenecks, inefficient use of cell resources, and increased metabolic burden on individual cells. An emerging area of focus has been on the use of cell co-cultures or small consortia wherein individual populations work together to accomplish a desired output in cooperation with the rest of the consortia[9–15]. There are many potential advantages to using multi-cell systems over traditional clonal populations including the potential for division of labor and reduced metabolic burden on individual strains, ability for specialization and ease of optimization, and options for plug and play[9–11]. While promising, the use of co-cultures requires not only regulation of gene transcription within each population, but also regulation of each cell population within the consortia.

Relatively few studies have been devoted to developing devices or systems that regulate the compositions of subpopulations within consortia. Often, studies that use multi-cell populations to carry out a coordinated task, such as producing biofuels or chemicals, rely on specific inoculation ratios or similar manual strategies to optimize the ratio of each population[16–18]. Alternatively, microfluidic and other devices can modulate the relative contributions of subpopulations by providing means to sequester or retain one population relative to another (e.g., using immobilization strategies) or by fluidically, but not physically, connecting populations (e.g., via porous membranes or 3D-printed microenvironments[19]). A potentially more powerful approach that does not rely on equipment is to reengineer native cell-cell signaling systems in such a way as to enable the *autonomous* coordination of subpopulation densities. We and others have previously exploited quorum sensing (QS), a bacterial form of cell–cell communication, to engineer communication circuits amongst and between bacterial strains to coordinate behaviors[20–24] or enable density dependent activation of desired behavior[25,26]. QS circuits and signals have also been used to alter cell densities by, for instance, activating production of toxins or lysis genes in order to program stationary phase cell density of a monoculture[27] and to create co-cultures with defined behavior[28,29]. Similar strategies have been used to design co-cultures with a range of social interactions[30].

Here, we develop a platform for autonomous and targeted regulation of consortia composition based on the prevailing level of an environmental cue, autoinducer-2 (AI-2) (Fig. 1a). The universal QS signal, AI-2, which is recognized and produced by many species of bacteria[31,32], broadly indicates cell population density and is also likely to be an important signal in natural consortia or microbiomes[33]. Therefore, our synthetic system can be modulated based on an important signal often present in bacterial environments, AI-2, that is not easily measured on-line by users, either in fermentation reactions or in natural consortia. We achieve this by rewiring bacterial QS systems so that the growth rate of communicating consortia members is controlled by interspecies signaling. Thus, we present development of a signaling and control system that imparts trans-species communication and growth rate control. Our synthetic co-culture consists of an *E. coli* translator strain that senses AI-2 and translates this into an orthogonal QS signal (AI-1). This translator strain's output, in turn, mediates the growth rate of the second strain. That is, a second engineered *E. coli* controller strain has signal-mediated tunable growth rate, regulated by the level of the

second, species-specific autoinducer signal, AI-1. Thus, the translator population produces AI-1 after sensing AI-2, in turn regulating the growth rate of the AI-1 responsive controller strain and subsequently the composition of the synthetic consortia based on the prevailing AI-2 level (Fig. 1b).

There are two important and innovative aspects to our design. First, QS-mediated communication between subpopulations enables composition adjustment to occur autonomously. Importantly, the system is based on the prevailing concentration of a common naturally occurring autoinducer (AI-2) and the controller signal is based on an orthogonal species-specific autoinducer (AI-1) that has no function beyond its native host (*Pseudomonas aeruginosa*), a strain either included or not, based on system design. The second aspect of our design is signal-mediated tunable growth rate of bacteria via positive feedback. This is made possible by regulation of HPr, a phosphotransferase system (PTS) protein[34], important for sugar (including glucose) transport in bacteria. We recently discovered that transgene expression of HPr in isogenic null mutants enables accelerated growth[35]. By controlling HPr expression via QS signaling, we enable autonomous subpopulation control. Importantly, our strategy positively modulates cell growth rate, preserving enhanced metabolic function, rather than increasing cell death (e.g., through expression of toxins or lysis genes), a strategy previously used by others[27–29]. Regulating expression of a critical gene for methionine synthesis has also been used to regulate cell growth, although this strategy requires use of dropout media[36].

In this paper, we develop and characterize each construct of the synthetic co-culture and then demonstrate autonomous regulation of co-culture composition based on initial AI-2 levels in batch and extended batch conditions. We create a simple mathematical model of the autonomous consortia regulator and show that the model can be used to either target a specific population composition or predict co-culture behavior given specific inputs. The model can then be used to explore parameter ranges and synthetic biology designs for future applications.

## Results

**AI-1 signal controlled cell growth rate.** We first tested *E. coli* cell growth rate control through transcriptional regulation of *ptsH*, a gene involved in sugar transport. HPr (encoded by *ptsH*) is widely recognized as one of a series of proteins (e.g., E1, HPr, EII) that sequentially transfers a phosphoryl group from phosphoenolpyruvate (PEP) to glucose (or other PTS carbohydrate)[34]. HPr is highly conserved[37]. We recently discovered that HPr interacts with the AI-2 kinase LsrK, influencing AI-2 uptake[35]. We leveraged the AI-2 modulating functionality later when constructing the AI-2 sensing cell. Here, we demonstrated that *ptsH* mutant strains grow more slowly than wild type strains in minimal media containing glucose (Supplementary Fig. 1a). When *ptsH* was placed under an IPTG inducible promoter in a *ptsH* mutant, growth rate could be controlled based on IPTG addition (Supplementary Fig. 1b). Importantly, we demonstrated that inducing expression of *ptsH* results in an increase in cell growth rate. Equally importantly, this behavior is irrespective of whether or not the media contains glucose as the principal carbon source (Supplementary Fig. 1c).

Based on this proof of concept, we next engineered QS signal controlled growth rate. To construct the controller strain, *ptsH* was placed under control of the AI-1 *lasI* QS promoter on the plasmid pAHL-HPr (Fig. 2a) in a *ptsH* mutant strain PH04. Elsewhere on the plasmid both dsRedExpress2, for cell visualization, and LasR, required for *lasI* promoter activation, were expressed under a constitutive T5 promoter. Addition of AI-1 to the controller strain increased cell growth rate up to 1.8 times the

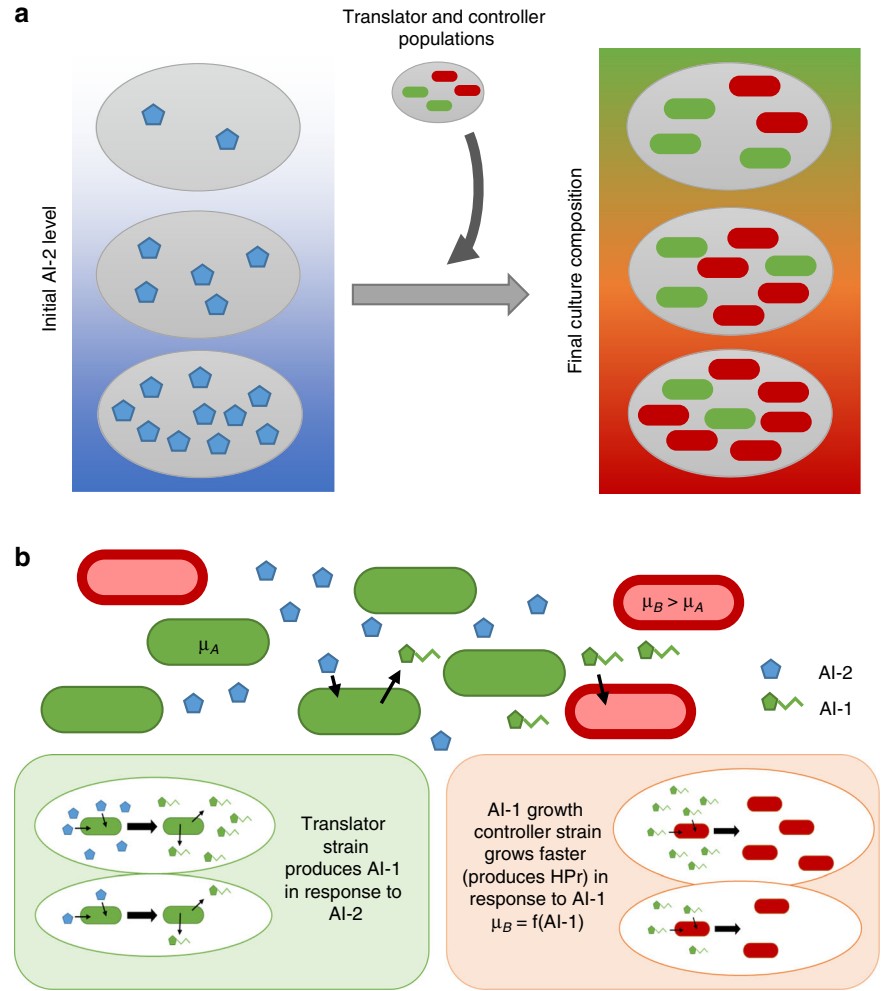

**Fig. 1** Design of autonomously regulated co-culture. **a** Engineered co-culture with AI-2 regulated composition. When the translator and controller populations are added to environments with AI-2, the resulting culture composition varies based on the AI-2 level. **b** Depiction of each strain in engineered co-culture. The translator strain senses AI-2 level and produces AI-1. The AI-1 growth controller strain produces HPr in response to AI-1, which alters cell growth rate and changes the culture composition

baseline growth rate in a dose dependent manner (Fig. 2b). The measured specific growth rate (between 1 and 4 h after AI-1 addition) served as the basis for a Monod-type model of growth rate based on the AI-1 level (Fig. 2c).

**AI-1 modulates composition in controller cell co-cultures**. The AI-1 regulated controller cells were then co-cultured with other strains to verify that addition of AI-1 altered the composition of the co-culture. Growth controller cells expressing a red fluorescent protein (PH04 pAHL-ptsH) were co-cultured with either PH04 pCT6 or TOP10 pT5G[7]. PH04 pCT6 has a similar baseline growth rate to PH04 pAHL-ptsH, while TOP10 pT5G grows significantly faster. Cultures were inoculated at approximately equal cell densities and supplemented with varied AI-1 levels. After 8 h, samples were taken for analysis using fluorescence microscopy and quantified using ImageJ. As expected, when controller cells were cultured with PH04, culture composition of the controller cells increased with increasing AI-1 concentration (Fig. 3a). A similar trend was seen when cells were cultured with TOP10, despite the faster growth rate of TOP10 (Fig. 3b). That is, in co-cultures without AI-1, the fraction of the controller cells in the TOP10 co-cultures actually dropped significantly from the initial ~0.5 to ~0.09 during the ensuing 8 h. Addition of AI-1

counteracted that growth rate difference and resulted in a higher level of controller cells during the same 8 h period. These results demonstrate that irrespective of other strains in the culture and their respective growth rates, AI-1 (which is not an *E. coli* QS signaling molecule) modulates the growth rate of the engineered controller strain and the change occurred sufficiently rapidly so as to enable an observable change in culture composition—notably even under relatively short term batch conditions (as opposed to fed-batch, repeated batch, or continuous cultures).

Next, the AI-1 regulated controller strain was co-cultured with cells that produce AI-1 when induced. PH04 pSox-LasI and PH04 pAHL-HPr were co-cultured together. Plasmid pSox-LasI contains *lasI*, which synthesizes AI-1, under the pyocyanin inducible *soxS* promoter[7]. In this experiment, the controller strain receives a signal from a translater strain that, in turn, produced AI-1 in response to pyocyanin. In this way, the co-culture control scheme is shown to respond to a particular molecular cue. Here, each culture was inoculated at approximately equal starting densities and co-cultures were either exposed or not, to pyocyanin. Exposed cultures showed increased production of AI-1 and corresponding increased composition of PH04 pAHL-HPr within 5 h (Fig. 3c). These results demonstrate that AI-1 produced from an alternate strain can modulate growth rate of the controller strain, and that this can occur on timescales required to affect

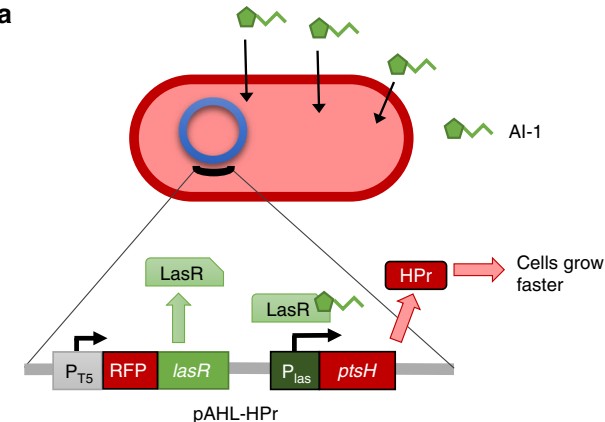

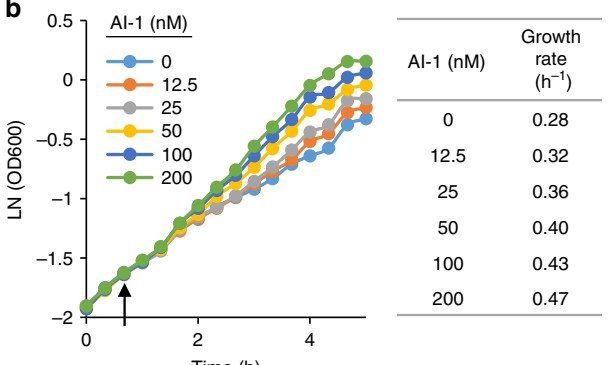

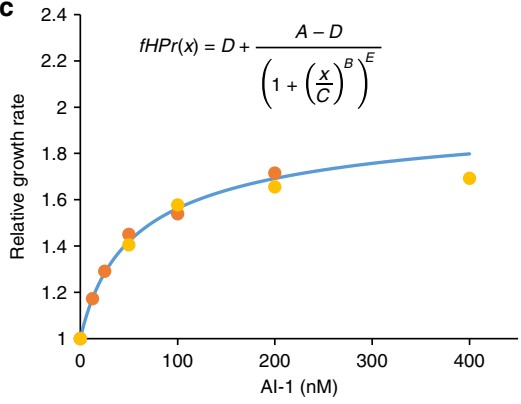

**Fig. 2** AI-1 quorum sensing signal controlled growth rate through expression of HPr. **a** Schematic of AI-1 growth responsive controller cells (PH04 pAHL-HPr). AI-1 binds LasR (constitutively expressed) and activates the *las* promoter resulting in expression of HPr which increases cell growth rate. **b** Growth curves of PH04 pAHL-HPr cultures grown with varying levels of AI-1. Cultures were grown to OD 0.15 ($t = 0$) and AI-1 was supplemented at 40 min (indicated by arrow). Table shows the average growth rate from 1 to 4 h after AI-1 addition. **c** Growth rate relative to basal growth rate (either 0.28 or 0.32 h$^{-1}$) for different AI-1 concentrations is plotted for two separate experiments (yellow and orange dots). Function *fHPr* describing relative growth rate as a function of AI-1 is plotted (blue line). $A = 1$, $B = 1$, $C = 29$, $D = 2.1$, $E = 0.48$. Source data are provided as a Source Data file

change in a batch process. In addition, this demonstrates potential feasibility of a user regulated co-culture based on a user-specific application of a molecule or inducer, in this case pyocyanin. We next engineered the translator strain to create an autonomously regulated co-culture based on the pervasive AI-2 signaling molecule.

**Design of AI-2 sensing translator cells.** To construct cell lines that sense AI-2 and produce AI-1, we engineered strains to activate expression of LasI, which synthesizes AI-1, when the AI-2 *lsr* promoter is activated. We used a two plasmid system to amplify expression from the weak *lsr* promoter[25] (Fig. 4a). Briefly, AI-2 is phosphorylated by LsrK and phosphorylated AI-2 relieves repression of the promoter by LsrR, increasing transcription of the *lsr* transporter genes and accelerating AI-2 uptake. At the same time, AI-2-mediated activation of the *lsr* promoter results in transcription of T7 RNA polymerase from plasmid pCT6 and subsequent transcription of *lasI* from plasmid pET-LasI.

The system was first constructed in the *E. coli* host strain CT104, which is a *luxS* mutant (e.g., incapable of producing AI-2). CT104 also lacks *lsrFG*, responsible for degrading the phosphorylated AI-2 signal, increasing sensitivity of the cell to AI-2[38]. We verified this cell line, CT104 pCT6 pET-LasI, produced AI-1 when cultured in conditioned media (CM) from AI-2 producing BL21 (Supplementary Fig. 2). BL21 were grown to varying cell densities, CM was collected, and CT104 cells were grown in the collected CM. Importantly, AI-1 produced by the CT104 cells correlated with the density of the AI-2 producing BL21 cells (Supplementary Fig. 2a). We also tested whether translator cells added to consortia with varying fractions of AI-2 producing cells could produce the AI-1 signal based on the consortia composition. CT104 cells were added to cultures of varying ratios of BL21 (*luxS*+) to BL21 Δ*luxS*. After 6 h, the level of the AI-1 signal in the extracellular media was indicative of the culture composition, which, in turn, is based primarily on the initial fraction of *luxS*+ cells[25] (Supplementary Fig. 2b).

The above experiments demonstrated that a cell line could be constructed to produce AI-1 in response to AI-2 from AI-2 producing cells. These experiments were performed in LB media and not in media with glucose. The presence of glucose inhibits AI-2 uptake and *lsr* promoter activity[39] and the use of the CT104 cells could potentially be limited to media without glucose (see Supplementary Fig. 3 for scheme of AI-2 QS pathway). Based on knowledge of the AI-2 QS system, we attempted to engineer a strain that was capable of AI-2 uptake and *lsr* promoter activation in glucose containing media. In this way, our results could be more generally applied. Previously, we demonstrated that HPr (encoded by *ptsH*) interacts with LsrK, inhibiting LsrK kinase activity, and that *ptsH* mutants have been shown to uptake AI-2 even in media containing glucose[35]. Hence, we hypothesized, use of PH04 (Δ*ptsH* Δ*luxS*) as a host strain would allow for AI-2 based production of AI-1 in media containing glucose. We tested the translator cells using both host strains in LB and M9 glucose media with and without AI-2. The level of AI-1 produced was dependent on addition of AI-2, but also on the host strain and the media (Fig. 4b). Both strains produced AI-1 when cells were added to LB media with AI-2. As expected, in M9 media, using the CT104 host strain resulted in a small or insignificant fold change in AI-1 activity when comparing cultures with and without AI-2. Importantly, the PH04 translator cells however showed AI-2 activated production of AI-1 in M9 + glucose media. In this way, the engineered system could be used in glucose-containing media.

**Characterization of PH04 translator cells.** PH04 translator cells uptake the AI-2 signal molecule, transduce the signal by activating expression of LasI and synthesize and secrete AI-1. The desired AI-1 output is then a function of the AI-2 input. We characterized AI-2 signaled production of AI-1 in the PH04 translator strain over time and for a range of biologically relevant AI-2 concentrations. The rate of AI-1 production by PH04 translator cells was found to be dose dependent on AI-2 (Fig. 5a).

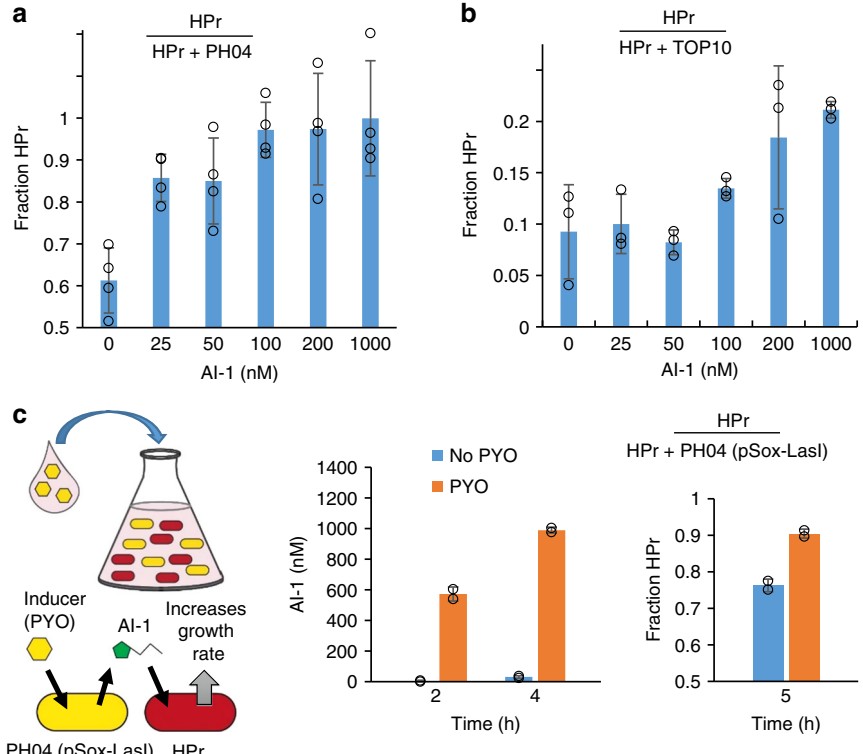

**Fig. 3** AI-1 regulates cell growth rate in co-cultures. **a** PH04 pAHL-HPr (abbreviated as HPr) co-cultured with PH04 pCT6 (abbreviated PH04). Each culture was inoculated 0.5% for an initial HPr fraction of ~0.5. Indicated AI-1 level was added during inoculation. Samples were collected for analysis of co-culture composition after 8 h. Error bars represent s.d. of technical quadruplicates. **b** PH04 pAHL-HPr (abbreviated as HPr) co-cultured with TOP10 pT5G (abbreviated as TOP10). Each culture was inoculated 0.5% for an initial HPr fraction of ~0.5. Indicated AI-1 level was added during inoculation. Samples were collected for analysis of co-culture composition after 8 h. Error bars represent s.d. of technical triplicates. **c** PH04 pAHL-HPr (abbreviated as HPr) co-cultured with PH04 pSox-LasI with or without 1 μM PYO induction. Initial fraction HPr in co-culture was ~0.5. Samples collected for AI-1 measurement after 2 h and 4 h and co-culture composition analysis after 5 h. Error bars represent s.d. of biological duplicates. Source data are provided as a Source Data file

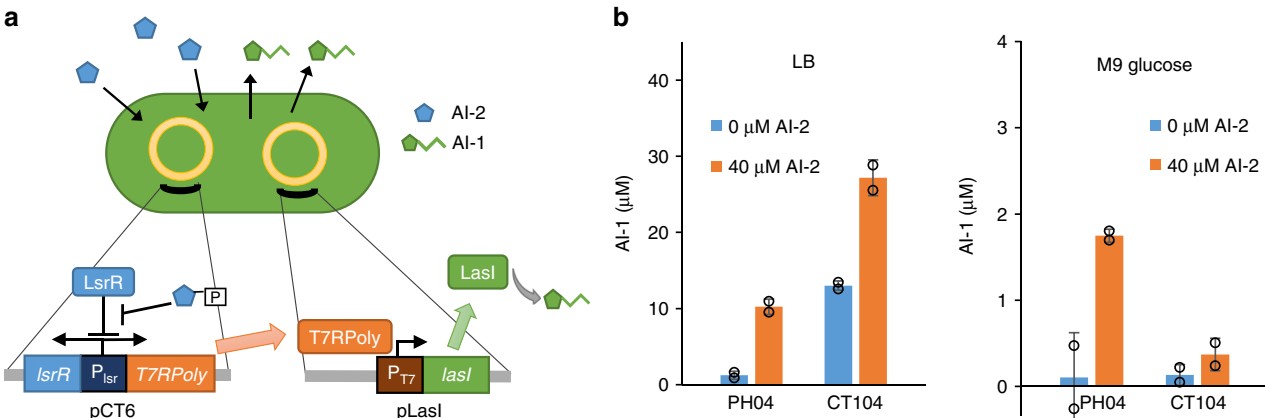

**Fig. 4** Engineered translator cells produce AI-1 in response to AI-2. **a** Translator cells couple the AI-2 and AI-1 quorum sensing circuits using a dual plasmid system. Phosphorylated AI-2 activates expression of T7 RNA polymerase and subsequent expression of LasI which synthesizes AI-1. **b** AI-1 produced by CT104 pCT6 pLasI or PH04 pCT6 pLasI grown in different media with and without AI-2 addition. AI-2 was added as indicated at ~OD 0.1 and samples for AI-1 measurement were taken 6 h later. Error bars represent s.d. between technical duplicates. Source data are provided as a Source Data file

We note that under these and similar conditions, AI-2 is mostly consumed within 4 h (Fig. 5b). Addition of AI-2 or AI-2 based production of AI-1 in these batch cultures resulted in no observable decrease in cell growth (Fig. 5c). We then characterized the rate of AI-1 production on a per cell basis over time for each tested AI-2 concentration by plotting a logistic function through the AI-1 data, determining the derivative of this equation over time, and dividing the derivative by the cell density over time (Supplementary Table 1) yielding a time-dependent specific production rate. The resulting plots (Fig. 5d) show the estimated per cell rate of AI-1 production as a function of added AI-2 and time from AI-2 addition. As will be shown later, this aligns with

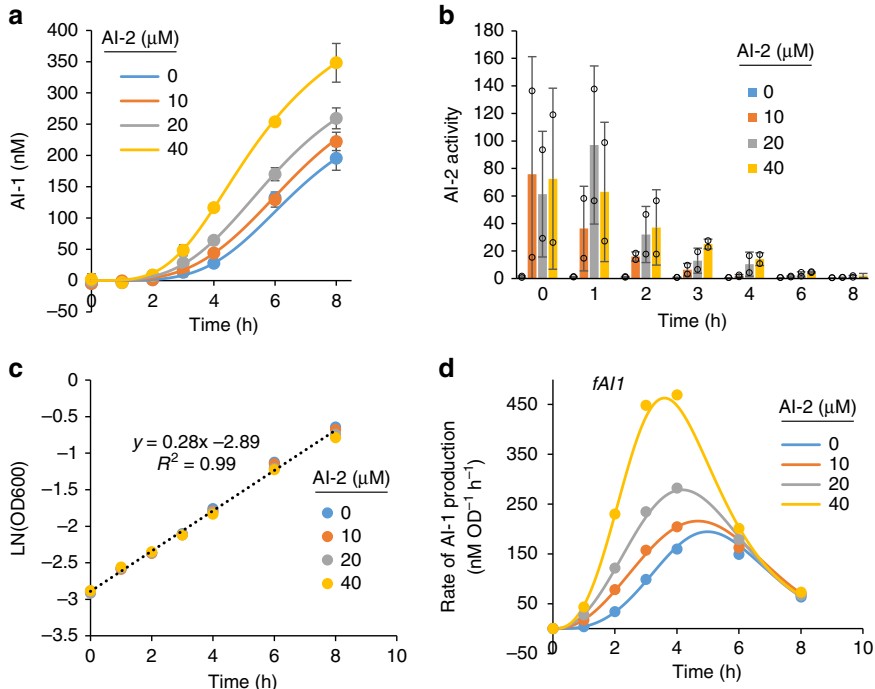

**Fig. 5** Characterization of AI-1 production in translator cells. PH04 translator cells cultured in M9 glucose media with varying concentrations of AI-2. Cells were inoculated from overnight cultures and AI-2 was added as indicated at $t = 0$. **a** Extracellular AI-1 levels over time. Error bars represent s.d. of technical duplicates. **b** Extracellular AI-2 activity over time. Error bars represent s.d. of technical duplicates. **c** Cell density over time. **d** Functions for AI-1 production rate, *fAI1*, for different AI-2 concentrations (solid lines) and AI-1 production rates calculated with experimental data (dots). Source data are provided as a Source Data file

an underlying observation we have observed that the trajectory of gene expression in response to an initial cue is fairly robust[21,38,40].

Next, the effect of AI-1 produced by the PH04 translator cells on the growth rate of the AI-1 responsive controller cells was tested. That is, growth responsive cells were added to CM containing AI-1 from translator cells that had been exposed to varying levels of AI-2 and cultivated for ~3 h (Supplementary Fig. 4a). Beyond what was shown in Fig. 5 where AI-1 is generated from direct exposure to AI-2, this experiment demonstrates that the cell translation of AI-2 into AI-1 can be done with the necessary expression and cell culture dynamics so as to influence the growth rate of the second population (Supplementary Fig. 4b). We note also, that the dynamic growth rate of the controller cells was shown to decrease in time over the ensuing 3 h. This observation was made more dramatic in later co-culture experiments.

**Autonomous regulation of co-culture composition**. We next added the translator cells (referred to as Population A) and the AI-1 responsive controller cells (Population B) to solutions having a range of initial AI-2 concentrations. In co-cultures of translator cells and controller cells, the translator cells autonomously regulate consortia composition based on initial AI-2 level. In Supplementary Fig. 5, the co-cultures with increased initial AI-2 levels resulted in an increase in AI-1 (Supplementary Fig. 5b) and a corresponding increase in relative abundance of Population B (Supplementary Fig. 5a). These results demonstrated the concept of signal-mediated autonomous control of consortia population. Importantly, estimates of growth rate for Population A and Population B agreed with expected growth rate values measured during monoculture experiments (Supplementary Fig. 5c).

Then, having demonstrated that translator cells could regulate consortia composition based on exposure to AI-2, we developed a mathematical model to characterize the system dynamics. In this way, one may predict co-culture behavior given specific initial conditions, growth rate designs, etc., in order to determine parameters required to target a desired output. This conceptually simple mathematical model was created to predict co-culture behavior using data from the individual strains. The model consists of four ordinary differential equations, one for each population density, substrate concentration, and AI-1 concentration (Supplementary Tables 2 and 3). Monod growth kinetics with a constant yield coefficient were used to model cell growth and substrate concentration, with additional functions accounting for the production and/or effects of the QS molecules. The AI-1 produced by Population A is based both on the level of AI-2 exposure and time (*fAI1*). In the previous work[38], we found that time-dependent trajectories of cell behavior were a consequence of initial exposure to autoinducer so that time-dependent functions of AI-1 production would be plausible here. The growth rate of Population B was formulated based on the prevailing level of AI-1 (*fHPr*). Maximum specific growth rates were measured using experimental data, yields were estimated based on experimentally observed stationary phase density and known initial substrate concentrations, and $K_1$ and $K_2$ values were chosen based on literature values for glucose. The MATLAB (Version R2016a) ode45 solver was used to solve the system of ODEs. The model can be used to show how a co-culture population is predicted to change with time given these simple phenomenological rate equations and best-fit constants. As noted, this provides the basis for determining whether a co-culture can even be predicted to evolve over the limited times available in a batch culture. Importantly, our results from monocultures are used to simulate experimental results from co-cultures.

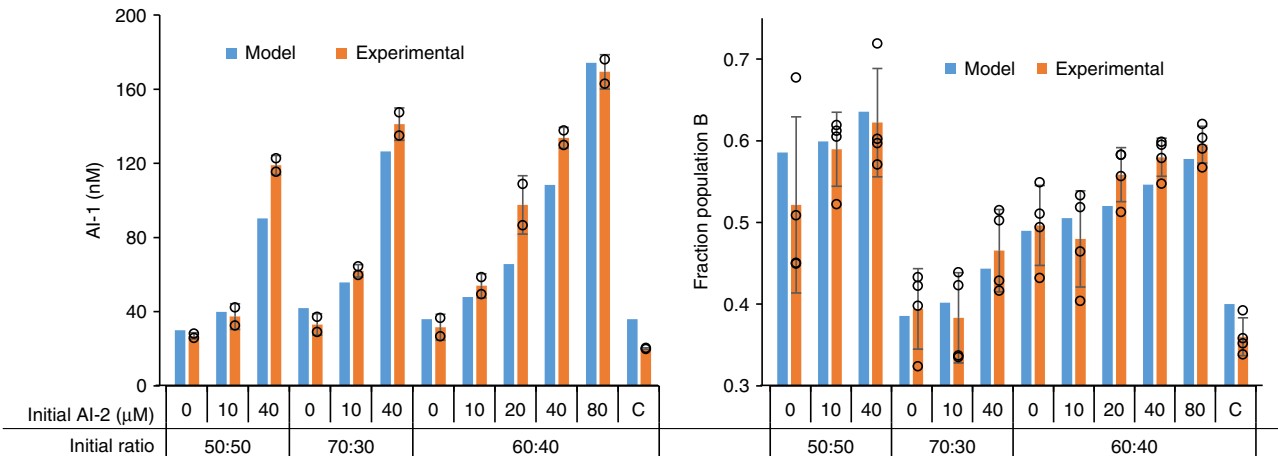

**Fig. 6** Predicting co-culture behavior using mathematical model. Co-cultures of A and B were added to media containing varied AI-2 concentrations and samples for AI-1 (left) and co-culture composition (right) measurement were collected after 5 h. Both cell lines were inoculated from overnight cultures to a combined initial OD of 0.05. The initial ratio of A:B is indicated. The control "C" used PH04 pAHL-sfGFP instead of PH04 pAHL-HPr as Population B and used media with 0 μM AI-2. Both model results and experimental results are shown. Error bars represent s.d. between technical duplicates (AI-1 measurements) and technical quadruplicates (composition measurements). Source data are provided as a Source Data file

That is, we next evaluated autonomously programmed co-culture control via model predictions. Co-cultures of Populations A and B were placed together for a range of initial cell populations and were then added to media with prescribed levels of AI-2 in order to set in motion a population trajectory. In our system, the initial composition is selected based on the ratio of cells supplied in the co-culture and then the culture composition is autonomously adjusted over time based on the AI-2 level in the media. We compared results to our model. In Fig. 6, the co-culture composition and AI-1 level after 5 h is shown for a variety of conditions including varied initial A:B ratios and AI-2 level. In these tests, we used one initial population cell density. Importantly, we found that our model predicted well the experimental results of both the AI-1 level and co-culture composition. We note also that the model can be used to select initial conditions that give a desired output. For instance, to target a Population B with relative cell density of 55% at 5 h, the co-culture could be started with an initial A:B ratio of 60:40 and AI-2 concentration of 40 μM. Other scenarios were tested and validated, as depicted. These results clearly demonstrate that the initial condition of cell composition (e.g., ratio of A:B) and the exposure to different levels of AI-2 both influence the trajectory of the co-culture population. Equally important, however, is that the orthogonal signal molecule, AI-1, behaved as modeled. We anticipate that, by extension, inclusion of this and other translator signals will enable further, varied or more complicated consortia to be designed and/or controlled.

That is, we tested the co-culture controller system by exposure to an AI-2 concentration, 80 μM, that was higher than the AI-2 concentrations used to characterize the translator cells, and for which we had no model. We used results of the co-culture (Fig. 6) to estimate behavior of translator cells in a monoculture at this AI-2 concentration. We then performed monoculture experiments by adding 80 μM AI-2 to translator cells and showed that we were able to predict monoculture behavior using the co-culture data and the model. Supplementary Fig. 6a shows the rate of AI-1 production predicted from the co-culture data and model (line) and the actual rate of AI-1 production during the monoculture experiment (data points). Supplementary Fig. 6b shows the predicted AI-1 levels in the monoculture over time compared to the actual measured AI-1 levels. The predicted AI-1

level was determined using the model, inputting in the estimated AI-1 production rate and the initial cell density of the monoculture.

Lastly, we tested our co-culture system over a more extended time period using repeated batch feeding with multiple AI-2 additions (Fig. 7). Here, our objective was to extend the population trajectory beyond that obtained by varying the initial composition and exposure to a fixed AI-2 level in a simple batch culture. In this way, we test the robustness of the control scheme. For example, in Fig. 6, for cultures initially at ~40% population B, we found we could only achieve levels of Population B that approached 60% and only by exposure to high levels (>80 μM) of AI-2. By testing a repeated batch system, we thought we could drive the B population to over 80% by simply resuspending and extending the system in time. We added our co-culture to media with varied levels of AI-2 and every 3 h we resuspended the cells in fresh media with additional AI-2. Immediately prior to each resuspension, samples were taken for analysis of AI-1 concentration and cell culture composition (Fig. 7a, b). Cell density and AI-2 activity were also measured (Supplementary Fig. 7). We found that in this more complex experimental set-up, the system generally worked as designed. We found that by exposure to lesser quantities of AI-2 (20 and 40 μM), we could reach nearly 80% population B. During this test, however, we found that our experimental results diverged from the results predicted by our mathematical model. We looked more closely at the AI-1 produced and the growth rate of the cultures during each 3 h segment in order to glean an understanding of the culture dynamics based on the observed divergence with the simple model. For instance, the AI-1 produced during the second and third 3 h cycles was higher than predicted by the model. In hindsight, this made sense because the cells, after the first batch cycle, had already produced LasI (which synthesizes AI-1), and the additional AI-2 was likely inducing further production of LasI (we did not include a degradation tag on LasI, so its maintenance should have been anticipated). We then adjusted the model so that the AI-1 production rate at the beginning of each subsequent resuspension in new media was the same as at the end of the previous cycle (Fig. 7c, solid lines). Incorporating these new functions for AI-1 production into the model resulted in values for AI-1 that closely fit the experimental AI-1 data (Fig. 7a, model

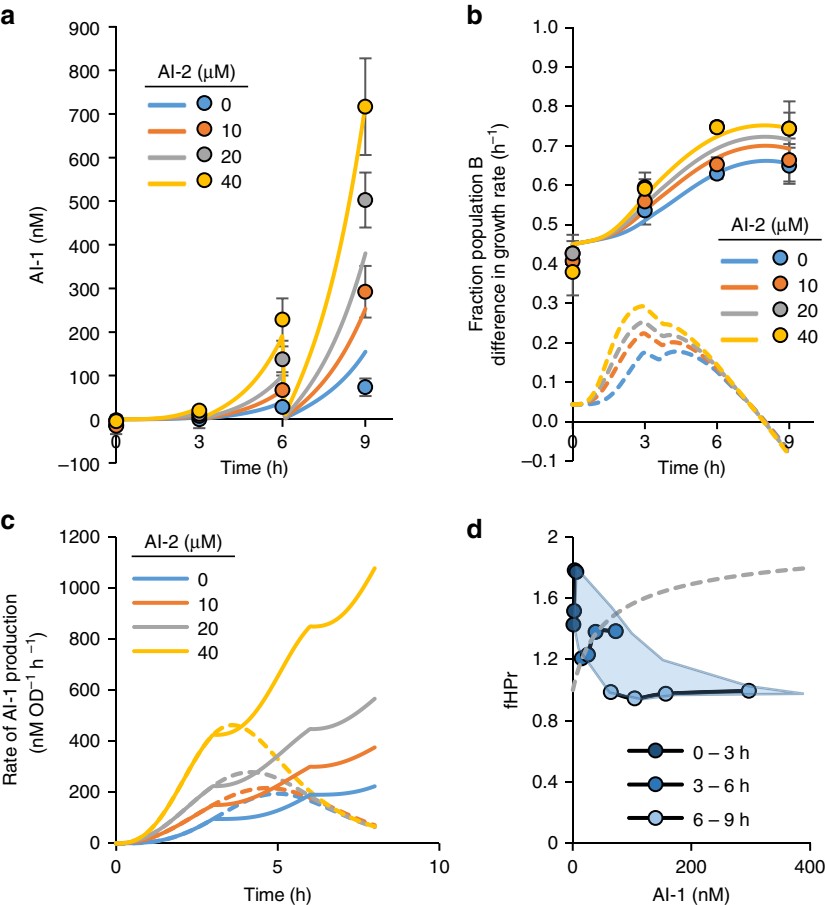

**Fig. 7** Co-culture system in repeated batch set-up. Co-cultures of A and B were inoculated to a total starting OD of ~0.05 in media with the indicated level of AI-2. Every 3 h, cultures were spun down and resuspended in fresh media with AI-2. Prior to resuspension samples were collected for AI-1 level and co-culture composition analysis. **a** AI-1 level in the cultures at inoculation ($t = 0$) and at the end of each 3 h segment. Data points show experimental data and lines represent adjusted model. Error bars represent s.d. between biological duplicates. **b** Fraction of Population B at inoculation ($t = 0$) and at the end of each 3 h segment. Data points show experimental data. Solid lines represent Fraction Population B predicted by adjusted model. Dashed lines represent the difference in growth rate between Populations A and B predicted by adjusted model. Error bars represent s.d. between biological duplicates. **c** Rate of AI-1 production over time for each AI-2 level (*fAI*) used in the original model (dashed lines) and the adjusted model (solid lines). **d** Data points are time averaged growth rate (Population B) versus time averaged AI-1 concentration for each 3 h segment and each AI-2 concentration. See Supplementary Note 1 for details on estimation of growth rate and AI-1. Dashed line shows original *fHPr* function. Source data are provided as a Source Data file

in solid lines). Similarly, we estimated the growth rate of the controller cells (see Supplementary Note 1) and found that the growth rate in combination with the AI-1 concentrations did not fit our earlier *fAI1* function (Fig. 7d). We note that to calculate the growth rate of the controller cells we assumed the growth rate of the translator cells remained constant, although they may have decreased slightly as a result of repeated AI-2 additions. While the controller growth rate seemed to initially increase as a function of AI-1, with subsequent resuspensions in fresh media the overall growth rate decreased. We had earlier noted a dynamic decrease in growth rate upon overexpression of HPr and LasI (Supplementary Figs. 4 and 5). Here, we suspect that either a metabolic burden placed on the cells from repeated exposure to high levels of AI-1 or reduced substrate or nutrient levels could have been causes for decreased growth during the later cycles. Importantly, by adjusting our model so that the effect of AI-1 on growth rate decreased with time (see Supplementary Note 2), we were able to fit our experimental data (Fig. 7b, adjusted model in solid lines) without adding model complexity.

In sum, our co-culture system responded as designed whether it was placed in media without AI-2 or with a high level of AI-2.

Moreover, our results showed controllable population densities that spanned 40 to 80% of the controller cells. Also, comparison to our original model gave insight into how the system behaved in this more complex experimental set-up; the translator cells appeared to produce higher levels of AI-1 over time while the controller cells seemed to have reduced AI-1 regulated changes in growth rate over time.

Finally, the model might provide a basis to explore in silico how the strategies used here for autonomously regulated cultures (marked by signal-regulated growth rate and native cell–cell signaling) could be extended to other systems, including user-regulated or programmable systems that could be dynamically controlled. For example, chemostat cultures are distinguished from the current autonomous system in that their outputs are directed by user-specified inputs, such as the dilution rate. As a first pass, we performed simulations of chemostat grown co-cultures by the addition of the standard flow terms to the batch model (Supplementary Fig. 8, Supplementary Tables 4 and 5). We found in some cases, that the dilution rate defines a steady-state culture composition which, in turn, can subsequently be "tuned" within a range constrained by the dilution rate. The "tuning" can

be achieved by externally modulating the cell–cell signaling, for example, by exogenous addition of signal molecules. These cases illustrate how the interplay between our autonomous system and other user-controlled systems might lead to more complex population trajectories. Our simulation results here serve as a conceptual framework for controlling consortia composition in more complex, user-guided systems. Further discussion of the chemostat simulations are in Supplementary Note 3.

## Discussion

Rapid advances in recombinant DNA technologies have greatly improved the ease of constructing engineered cells for applications ranging from bioprocessing (for production of valuable products) to smart bacteria capable of carrying out a multitude of functions. The bottleneck to further advancing these systems is typically not re-engineering genomes or their regulation, but optimizing cells to efficiently overproduce many proteins or carry out many functions without becoming metabolically over-burdened. To work around this, many have proposed using consortia, where tasks can be divided and cells can be specialized. We and others have designed systems using QS circuits to regulate or coordinate gene expression amongst populations or between subpopulations in small consortia. However, by designing QS signal-regulated growth rate and an orthogonal translator controller, we create a tool for an additional layer of control. That is, we enable regulation of composition of the co-culture or small consortia.

In our co-culture system, the composition of each subpopulation is autonomously regulated based on the level of AI-2 in the environment. The translator cells detect the level of AI-2 and produce the species specific signal molecule AI-1. The AI-1 regulated controller cells then adjust their growth rate based on the AI-1 signal level, which is a function of the initial AI-2 level. The result is an altered co-culture composition based on a native environmental cue, AI-2. We envision our system could be further engineered so that each population carries out part of a concerted effort or function that is autonomously fine-tuned when cells are placed into a particular environment. For example, we had previously reported on sensing cell networks in which the fractional level of responder cells indicates the previous environment they had surveyed[21]. Importantly, because of the partitioning of metabolic functions among subpopulations, our system reduces the potential metabolic burden on the cells—this done through use of QS to enable crosstalk between cells. Equally importantly, we have designed the system so that the AI-1 increases cell growth rate of the translator cells through increased transcription of a sugar transport protein *ptsH*, instead of by causing a reduced cell growth rate—the latter which may strain other engineered functions. We further believe using *ptsH* to regulate growth rate is likely a generalizable strategy due to its being well-conserved[37]. To estimate the behavior of a system using different strains or species, we suggest that prior to implementing the genetic circuit for growth control, one quantifies independently the growth characteristics of the individual strains. This information could be integrated into a simple model such as the one shown here, providing a range of culture dynamics that might be achieved. That is, a co-culture where the maximum growth rates are similar will be dramatically different than if they are very different, irrespective of our cell-based growth-controller module.

In summary, we believe this system can easily be adjusted for further application. In this work, we described a scenario where the system is used to respond to external levels of AI-2 produced by cells in an environment of interest. As an alternative, the system could be rewired so that either population produces AI-2.

In this case, both populations would grow naturally until a certain time when a threshold of AI-2 has been reached, at which time the growth rate of Population B would be signaled to change. That is, we believe the autonomous platform shown here, which functions independently and is accompanied by a simple model, could be used to design co-culture systems that allow for self-regulation of the composition of each subpopulation in multiple ways with regulation that requires no user or device intervention. Also by extension, our simple chemostat model predictions suggest we could maintain co-culture compositions at various steady states, but this would occur only with the interjection of well-defined user input (e.g., dilution rate, autoinducer addition, etc.). With such systems, or by inclusion as a subsystem within more complex environments, we expect to enable more widespread use of co-cultures—and the realization of the advantages that come with co-cultures or consortia in synthetic biology or metabolic engineering applications.

## Methods

**Strains and plasmids**. All strains, plasmids, and primers used are listed in Supplementary Tables 6 and 7. pTac-HPr was cloned for IPTG inducible expression of *ptsH*. The *tac* promoter with *ptsH* was amplified from pSkunk-HPr[35] using primers TacProm-PvuI-F and HPr-SpeI-R to add PvuI and SpeI restriction digestion sites upstream of the promoter and downstream of the stop codon, respectively. The fragment was inserted into pLSR[41], a vector containing the repressor for the *tac* promoter *lacI*, using restriction digestion with SpeI and PvuI.

To clone pAHL-HPr, sfGFP was replaced with *ptsH* in a previously cloned AI-1 fluorescent reporter plasmid (pAHL-Reporter_Red-Green) with constitutive expression of dsRedExpress2[42]. Primers HPr-SpeI-F and HPr-SacI-R were used to amplify *ptsH* while adding SpeI and SacI restriction digestion sites upstream and downstream of the start and stop codons, respectively. Restriction digestion of both the fragment and reporter plasmid followed by ligation were used to insert *ptsH* under the *lasI* promoter.

pSox-LasI, for PYO induced AI-1 production, was cloned by inserting the *lasI* gene with an ssRA degradation tag (AANDENYALAA) in place of the reporter *philov* in plasmid pTT01[7] using Gibson Assembly (New England Biolabs).

For the translator strain, pLasI was cloned using the Invitrogen Champion™ pET200 Directional TOPO® Expression Kit to insert *lasI* under the T7 promoter. Primers LasI_F and LasI_R were used to amplify *lasI* from the genome of PA01. Constitutive eGFP expression was added elsewhere on the plasmid, at the NruI restriction digestion site, although eGFP expression was not measured during experiments due to low intensity.

Strains PH03 and PH04 were derived from PH01 and PH02[35], respectively. Briefly, the antibiotic resistance cassettes in PH01 and PH02 were removed using plasmid pCP20[43] to create PH03 and PH04.

**Cell culture conditions**. For all experiments, cells were cultured at 37 °C and 250 rpm shaking. M9 media was prepared with 0.8% glucose and 0.2% casamino acids. Either M9 media or LB media was used for experiments as indicated. LB media was also used for cloning and to grow overnight cultures. Antibiotics were added based on resistance in plasmids. Concentrations used were 50 µg mL⁻¹ kanamycin (pLasI), 50 µg mL⁻¹ ampicillin (pCT6, pAHL-HPr, pSox-LasI), or 32 µg mL⁻¹ chloramphenicol (pTac-HPr). For multi-population experiments, antibiotics were added only if all populations contained genes for resistance. For instance, co-cultures of Population A and Population B contained ampicillin and not kanamycin. AI-1 (N-3-oxo-dodecanoyl-L-homoserine lactone) was purchased from Cayman Chemicals. AI-2 was chemically synthesized and generously provided by Dr. Sintim[44].

**Quorum sensing signal activity assays**. Bioluminescent reporter assays were used to determine AI-1 and AI-2 activity. Experimental CM samples were prepared by filtering supernatant through a 0.2 µM filter and storing at −20 °C until needed for activity assays. For the AI-1 activity assay[42], *E. coli* luminescent reporter cells containing plasmid pAL105[45] were grown in LB media overnight. In the morning they were diluted 2500 fold in LB media with 50 µg mL⁻¹ tetracycline and 50 µg mL⁻¹ kanamycin. Experimental samples were diluted in LB in order to be within the linear range of the assay. Samples for a standard curve of known AI-1 concentrations ranging from 0–60 nM AI-1 in LB were also prepared. Ten microliter of the experimental or standard curve samples were added to 90 µL of the reporter cells. Cultures were grown at 30 °C and 250 rpm shaking, and luminescent values were recorded after 3 h using a GloMax®-Multi Jr (Promega, Madison, WI, USA). Values were normalized to the negative control (fresh LB media with 0 µM AI-1). Each sample was performed in duplicate. The resulting standard curve along with the dilution factor of the sample were used to estimate AI-1 concentration in the original sample.

*Vibrio harveyi* BB170 was used to measure AI-2 activity[46]. *Vibrio harveyi* BB170 were grown overnight at 30 °C in AB media and diluted the next morning 5000 times in AB media with 10 μg mL$^{-1}$ kanamycin. Twenty microliter of CM experimental samples were added to 180 μL reporter cells and cultured at 30 °C and 250 rpm shaking. At 3 h and every half hour thereafter, luminescence values were recorded until the negative control reached a minimum luminescent value. Values were normalized to the negative control (fresh media with 0 μM AI-2). Each sample was performed in duplicate.

**Microscopy image analysis.** Microscopy images and ImageJ software were used to estimate fractions of each population within the co-cultures. An Olympus BX60 fluorescence microscope with a 20× objective lens and CellSans software were used for imaging cultures. For each sample, at four different locations or frames on the slide a bright field image and image with the fluorescent dsRed filter were taken. Fiji ImageJ software was used to count the cell numbers for each picture. For bright field images, the background was first subtracted using the "subtract rolling background" feature. Thresholds were set for each image type (bright field or fluorescent) and kept consistent for each day's experiment. For each frame, the red cell count was divided by the total cell count. For each sample, the values calculated for each of the four frames were averaged. Finally, a standard curve, where known amounts of each cell type were mixed directly before taking microscopy images was used to calculate the reported "Fraction Population B" value (Supplementary Fig. 9).

**Reporting summary.** Further information on research design is available in the Nature Research Reporting Summary linked to this article.

## Data availability
The data that support the findings of this study are available from the corresponding author upon request. The source data underlying Figs. 2–7 and Supplementary Figs. 1–2, 4–7, and 9 are provided as a Source Data file. All other revelant data, including plasmid sequences and plasmids, are additionally available upon request.

## Code availability
MATLAB Version R2016a or Simulink Version 8.7 (R2016a) were used to develop and solve the mathematical models. The MATLAB code for the batch and extended batch models are provided in Supplementary Software 1 and 2, respectively. The Simulink models used in this study are available from the corresponding author upon request.

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

## Acknowledgements

This work was partially supported by DTRA (HDTRA1-13-0037), NSF (DMREF #1435957, ECCS#1807604, CBET#1805274), and the National Institutes of Health (R21EB024102).

## Author contributions

K.S., C.Y.T., P.H., and W.E.B. developed the ideas. K.S. and P.H. designed and genetically engineered plasmids and strains. K.S. designed experiments. K.S. and M.P. performed experiments. K.S. developed and simulated the model in MATLAB and Simulink. K.S. and W.E.B. analyzed data and wrote the paper. W.E.B. supervised the work.

## Additional information

**Competing interests:** The authors declare no competing interests.

