## [Peer Review File · Nature Communications]

Editorial Note: Parts of this peer review file have been redacted as indicated to remove third-party material where no permission to publish could be obtained.

Reviewers' comments:

Reviewer #1 (Remarks to the Author):

I found the describe system linking AI-1 and AI-2 with Hpr to be extremely thorough and well described. I thought all of the figures were done very well and the rationale for the development of the model and the revision of the model was explained clearly.

When I began reading the paper, my immediate thought was that if the system were deployed in a continuous culture such as the chemostat suggested in the conclusion, it would truly be worthy of Nature Communications, illustrating the true impact of this elegant work. I anticipated that one of the later figures would show such an application. I believe this is the one missing piece for this to be a truly impactful work.

Reviewer #2 (Remarks to the Author):

The manuscript by Stephens et al reports a system that enable the control of the ratio of two populations in co-culture via a gene circuit that responds without user intervention. The authors have approached the problem in a novel way by developing a mechanism based on two types of cells, one that senses overall density and produces an orthogonal quorum molecule and a second that has its growth rate regulated by the quorum molecule produced. The authors have engineered this control to be via the expression of a sugar transporter, such that growth is enhanced by the production of the quorum molecule. The design here is unique and to my knowledge has not been proposed by others who tend to focus on engineering interdependence via auxotrophy (less controllable than the system proposed here) or via negative regulation, e.g. through toxin production (provides evolutionary pressure to escape). The system here has advantages for use in biotechnology and will be of broad interest to those working in synthetic biology as the need for engineering microbial consortia to perform tasks becomes more pressing. The authors also include a mathematical model that would allow users to design a system to achieve a particular composition of the population when desired.

The experiments are carefully conducted with each part of the system characterised before assembly into the final system. The figures and data are discussed clearly in the manuscript text and the figures themselves are clear and easy to interpret. The conclusions are supported by the results shown.

The main disadvantage of the system as proposed is that the ratio of the two populations changes with time over the batch culture as shown in Supplementary Figure 7 (and to some extent Figure 7). For a biosynthetic pathway split over two organisms, this seems like it could cause some issues with balancing fluxes. The authors do acknowledge that the ratio changes in the discussion and suggest that the use of continuous cultures in chemostats might solve the problem. However, perhaps a little more discussion would be useful here.

One other point that I think could be discussed further relates to the mathematical model. How much experimentation would be necessary for this to be used with different strains, i.e. which parameters would need to be remeasured?

-Supplementary Figures 1 and 4 do not state the number and type of replicates and does not show the variation between repeats of the experiment (either as individual points or using error bars)

There are a few minor typographical errors:

-Line 145 "Addition of AI-1 resulted counteracted", delete resulted

-Line 213, " The rate of AI-1 produced", should be " The rate of AI-1 production"

Overall, I found the work exciting and interesting and believe this to be a very novel system design for engineering microbial consortia

Stephens et al – Response to Reviewers

Reviewers' comments:

Reviewer #1 (Remarks to the Author):

I found the describe system linking AI-1 and AI-2 with Hpr to be extremely thorough and well described. I thought all of the figures were done very well and the rationale for the development of the model and the revision of the model was explained clearly.

When I began reading the paper, my immediate thought was that if the system were deployed in a continuous culture such as the chemostat suggested in the conclusion, it would truly be worthy of Nature Communications, illustrating the true impact of this elegant work. I anticipated that one of the later figures would show such an application. I believe this is the one missing piece for this to be a truly impactful work.

We like the idea of designing a system that could maintain both populations in a chemostat. However, we believe the extension of our work to include chemostat cultures is not trivial; it would actually drive the autonomous cell-cell signaling concepts that underpin our innovation towards a new user-defined direction. After careful consideration, we therefore have not incorporated chemostat experiments in the revised manuscript. We have included detailed reasoning at the end of this document in a response that we had previously provided to the Editor. We did, however, modify the manuscript significantly so that the autonomous designs shown here could be integrated into a system that would operate in a chemostat mode.

That is, we have modified our original model to investigate continuous cultures *in silico*. The results are very interesting, so much so, that we included the adapted model and subsequent simulations in new Supplementary information (Supplementary Figure 8, Supplementary Tables 4 and 5, and Supplementary Note 3). We have provided discussion of these simulations in the main text starting at the bottom of page 7. We have also modified the last paragraph of the discussion in reference to the chemostat simulations (page 9).

Reviewer #2 (Remarks to the Author):

The manuscript by Stephens et al reports a system that enable the control of the ratio of two populations in co-culture via a gene circuit that responds without user intervention. The authors have approached the problem in a novel way by developing a mechanism based on two types of cells, one that senses overall density and produces an orthogonal quorum molecule and a second that has its growth rate regulated by the quorum molecule produced. The authors have engineered this control to be via the expression of a sugar transporter, such that growth is enhanced by the production of the quorum molecule. The design here is unique and to my knowledge has not been proposed by others who tend to focus on engineering interdependence via auxotrophy (less controllable than the system proposed here) or via negative regulation, e.g. through toxin production (provides evolutionary pressure to escape). The system here has advantages for use in biotechnology and will be of broad interest to those working in synthetic biology as the need for engineering microbial consortia to perform tasks becomes more pressing. The authors also include a mathematical model that would allow users to design a system to achieve a particular composition of the population when desired.

The experiments are carefully conducted with each part of the system characterised before assembly into the final system. The figures and data are discussed clearly in the manuscript text and the figures themselves are clear and easy to interpret. The conclusions are supported by the results shown.

The main disadvantage of the system as proposed is that the ratio of the two populations changes with time over the batch culture as shown in Supplementary Figure 7 (and to some extent Figure 7). For a biosynthetic pathway split over two organisms, this seems like it could cause some issues with balancing fluxes. The authors do acknowledge that the ratio changes in the discussion and suggest that the use of

continuous cultures in chemostats might solve the problem. However, perhaps a little more discussion would be useful here.

We agree with the reviewer that it might be worthwhile having the autonomous control scheme operate in the opposite mode as shown here, by maintaining a constant ratio based on the prevailing conditions (e.g., a split biosynthetic pathway that involves a secreted intermediate that might independently alter the growth of either strain). We have added a discussion including this scenario (maintaining constant population ratios) and that of the chemostat (suggested by Reviewer #1) in an extension of our modeling efforts.

One other point that I think could be discussed further relates to the mathematical model. How much experimentation would be necessary for this to be used with different strains, i.e. which parameters would need to be remeasured?

The parameters that likely have the most influence on the dynamic range of the system are the growth characteristics of each strain. We believe a simple model, such as the one shown here, could then be used to estimate the boundaries of the system's behavior. For instance, when transferring the AI-1 growth "controller" module to a new strain, one would first need to construct a *ptsH* mutant-derivate strain. In this case, the growth rate of the parent strain could be used as an estimate of the upper limit of the growth rate of the engineered strain (when AI-1 is added). In this way, one could gauge the range of system behavior using the model. Also, in the split pathway case noted above, one would test whether secreted intermediates impacted the growth of either strain. These experiments are simple and the data would be relatively easy to incorporate into a model. Simulations and predictions would then be followed by the more time consuming process of constructing the engineered strain with the appropriate genetic regulatory structures. We incorporated some of this discussion into the manuscript on page 8.

-Supplementary Figures 1 and 4 do not state the number and type of replicates and does not show the variation between repeats of the experiment (either as individual points or using error bars)
This has been corrected.

There are a few minor typographical errors:

-Line 145 "Addition of AI-1 resulted counteracted", delete resulted

-Line 213, " The rate of AI-1 produced", should be " The rate of AI-1 production"

These have been corrected.

Overall, I found the work exciting and interesting and believe this to be a very novel system design for engineering microbial consortia.

Other changes made to the manuscript:

Additional minor changes were made to the manuscript to meet the requirements of the checklist. These include shortening the title, sub-heading titles, and text to meet the word or character limits. We also changed unit dimension notations, abbreviations, and notations for referencing supplementary material to meet the checklist requirements.

After careful review of our manuscript, we also modified Figures 3a and 3b. Throughout the manuscript, for all reported composition (fraction Population B) measurements, we had multiplied the values we obtained from the microscopy images and ImageJ analysis by a factor of 1.18 based on our standard curve using co-cultures of known compositions (see methods and Supplementary Figure 9). The 1.18 factor accounted for the fact that our method of analysis of culture composition slightly undercounted the percentage of red fluorescent cells. We mistakenly did not scale the data in Figures 3a and 3b by 1.18 in the initial submission (it had been used in all other datasets). We have corrected this in this revision. The result is that all the values (including the negative controls with 0 nM AI-1) are shifted upwards. The trends, data analyses, and conclusions drawn remain unaffected.

Stephens et al – Extension to Chemostat Cultures

We provide the following discussion concerning the fundamental differences between the chemostat culture suggested by the first reviewer and the repeated batch or extended batch culture that we have included in our original manuscript. The main points revolve around the following issues:

- Chemostats (i) require a user-specified dilution rate, (ii) operate with growth-limiting nutrient condition, and (iii) apply strong selective pressure leading to the predominance of cells having the highest growth rate.
- Our work is predicated on autonomous control.

In our system, cells detect an AI-2 level and autonomously change culture composition without user intervention. A chemostat, while attractive from a productivity point of view, would require user intervention, by design. That is, because of the strong selective pressure placed on mixed cultures, the slower growing cells would need to be engineered to grow faster than the faster growing cells at a given dilution rate. Also, the faster growing cells would need a trigger to slow their growth. At least one exogenous signal would be required to elicit these changes. The signal is unlikely to come from autoinducers secreted by cells as the level of the autoinducer production is a consequence of the growth (i.e., dilution rate), not population density. If the autoinducer production rate is not sufficient, it too will wash out.

The addition of the dilution rate essentially removes a degree of freedom and adds complexity. It also precludes complete autonomy. We like the idea, in general, but it is a very different problem.

To illustrate, we actually ran chemostat cultures years ago (DeLisa et al., *J. Bacteriol*, 2001). In the plots at the right, AI-2 production is a function of dilution rate (and growth rate). The more carbon flux through the synthesis pathways yielded more secreted autoinducer. Interestingly, the increase in AI-2 corresponded to increased growth independently from cell density (cell OD was essentially constant throughout, see Figure 1 from DeLisa et al. *J Bacteriol*, 2001). This observation complicates the design of a system where autoinducers reportedly signal population density for autonomous regulation of population level.

[Redacted]

Figure 1 AI-2 activity (a) and rate of AI-2 production (b) during steady-state transitions achieved by incremental upshift in culture growth rate (from 0.10 to 1.25 h⁻¹) of W3110/pHIL-2 chemostat cultures. The AI-2 production rate is the activity times the dilution rate, D (D = reactor volume/flow rate). Steady state was achieved in ca. 3 to 5 residence times. Increased growth rate was implemented by increasing the feed rate of fresh LB medium–50 mM glucose. Replicate samples agreed within 15%. Cell density changed less than 14% over the range of tested dilutions.
Figure from DeLisa, M.P., Valdes, J.J. & Bentley, W.E. Mapping stress-induced changes in autoinducer AI-2 production in chemostat-cultivated *Escherichia coli* K-12. *Journal of bacteriology* **183**, 2918-2928 (2001).

Instead of attempting to develop new genetic switches that enable user-specified input, we prefer to modify the manuscript to include further discussion of these chemostat ideas without incorporating new chemostat data. We will use our simple mathematical model to illustrate how aspects of our design could be applied towards a user-regulated system for a chemostat and which model parameters would be important to consider. At the same time we can include further discussion of how our system could be used with other host strains, including which parameters would be important to measure experimentally, as suggested by the second reviewer.

REVIEWERS' COMMENTS:

Reviewer #1 (Remarks to the Author):

I find the response to the chemostat question to be sufficient and reasonable. I appreciate the thoughtfulness of the response and the creation of content that allows this to be incorporated into such experiments in the future.

As noted in my original review, this is a strong contribution.

Reviewer #2 (Remarks to the Author):

The response letter and the changes made to the manuscript address the comments raised by me and the other reviewer. The authors decided against an experimental implementation of the chemostat, but do include some modelling results in the supplementary information. Overall, I understand their reasoning for doing so and agree with them that it would be a substantial amount of work.

The paper is novel and interesting and makes an excellent contribution to synthetic biology.

The authors have helpfully provided the numerical data sets used to create the figures as a supplementary file. I wonder if they would also consider providing the DNA sequences of the constructs.

Stephens et al – Response to Reviewers

Reviewers' comments (after second submission):

Reviewer #1 (Remarks to the Author):

I find the response to the chemostat question to be sufficient and reasonable. I appreciate the thoughtfulness of the response and the creation of content that allows this to be incorporated into such experiments in the future.

As noted in my original review, this is a strong contribution.

Reviewer #2 (Remarks to the Author):

The response letter and the changes made to the manuscript address the comments raised by me and the other reviewer. The authors decided against an experimental implementation of the chemostat, but do include some modelling results in the supplementary information. Overall, I understand their reasoning for doing so and agree with them that it would be a substantial amount of work.

The paper is novel and interesting and makes an excellent contribution to synthetic biology.

The authors have helpfully provided the numerical data sets used to create the figures as a supplementary file. I wonder if they would also consider providing the DNA sequences of the constructs.

We thank the reviewers for their comments. In reference to the request from Reviewer #2 to consider including DNA sequences of the constructs, we have decided not to include the DNA sequences owing to the restrictions on file types. Common file types used for annotating DNA sequences (.dna, .gb) are not listed as acceptable file types to include as supplementary material with submission to Nature Communications. We believe a text file of the plasmid sequences would not be helpful as it would not be annotated. However, the constructs and/or corresponding DNA sequences, will be available from the corresponding author upon request.